# Design, Synthesis, and Biological Evaluation of Novel Dihydropyridine and Pyridine Analogs as Potent Human Tissue Nonspecific Alkaline Phosphatase Inhibitors with Anticancer Activity: ROS and DNA Damage-Induced Apoptosis

**DOI:** 10.3390/molecules27196235

**Published:** 2022-09-22

**Authors:** Nazeer Ahmad Khan, Faisal Rashid, Muhammad Siraj Khan Jadoon, Saquib Jalil, Zulfiqar Ali Khan, Raha Orfali, Shagufta Perveen, Areej Al-Taweel, Jamshed Iqbal, Sohail Anjum Shahzad

**Affiliations:** 1Department of Chemistry, COMSATS University Islamabad, Abbottabad Campus, University Road, Abbottabad 22060, Pakistan; 2Centre for Advanced Drug Research, COMSATS University Islamabad, Abbottabad Campus, Abbottabad 22060, Pakistan; 3Department of Chemistry, Government College University, Faisalabad 38000, Pakistan; 4Department of Pharmacognosy, College of Pharmacy, King Saud University, P.O. Box 2457, Riyadh 11451, Saudi Arabia; 5Department of Chemistry, School of Computer, Mathematical and Natural Sciences, Morgan State University, Baltimore, MD 21251, USA

**Keywords:** dihydropyridines, pyridines, anticancer activity, apoptosis (flow cytometry; ROS; PI and DAPI), CDK4/6, alkaline phosphatase

## Abstract

Small molecules with nitrogen-containing scaffolds have gained much attention due to their biological importance in the development of new anticancer agents. The present paper reports the synthesis of a library of new dihydropyridine and pyridine analogs with diverse pharmacophores. All compounds were tested against the human tissue nonspecific alkaline phosphatase (*h*-TNAP) enzyme. Most of the compounds showed excellent enzyme inhibition against *h*-TNAP, having IC_50_ values ranging from 0.49 ± 0.025 to 8.8 ± 0.53 µM, which is multi-fold higher than that of the standard inhibitor (levamisole = 22.65 ± 1.60 µM) of the *h*-TNAP enzyme. Furthermore, an MTT assay was carried out to evaluate cytotoxicity against the HeLa and MCF-7 cancer cell lines. Among the analogs, the most potent dihydropyridine-based compound **4d** was selected to investigate pro-apoptotic behavior. The further analysis demonstrated that compound **4d** played a significant role in inducing apoptosis through multiple mechanisms, including overproduction of reactive oxygen species, mitochondrial dysfunction, DNA damaging, and arrest of the cell cycle at the G1 phase by inhibiting CDK4/6. The apoptosis-inducing effect of compound **4d** was studied through staining agents, microscopic, and flow cytometry techniques. Detailed structure–activity relationship (SAR) and molecular docking studies were carried out to identify the core structural features responsible for inhibiting the enzymatic activity of the *h*-TNAP enzyme. Moreover, fluorescence emission studies corroborated the binding interaction of compound **4d** with DNA through a fluorescence titration experiment.

## 1. Introduction

Alkaline phosphatase (AP) has been found to be a reliable serum tumor marker in the early detection of sarcomas, as such tumor markers are released into biological system in response to cancerous cells [1,2]. AP is deliberately used as a universal enzyme localized in entire body tissues, but their concentration is mostly high in some specific regions of body, such as in bones, the kidney, the placenta, and the liver [3,4,5]. An overexpression of alkaline phosphatases (APs) has a close relationship with different types of cancers [6,7,8]. In fact, a higher concentration of AP in blood serum is an initial indicator of bone lesions [9,10,11]. AP and its bone-specific isoenzyme AP may be the best tools for the detection of bone metastases in breast cancer [12]. The placental overexpression of APs is mostly observed in cells derived from the breast cancer region [13,14]. Similarly, higher concentrations of germ cell and intestinal APs were noticed in the case of choriocarcinoma and hepatocellular carcinoma, respectively [15]. In addition, breast cancer induces bone metastasis, and further progression results in liver cancer [14]. Liver cancer also arises due to the overproduction of intestinal alkaline phosphatases [16]. Alkaline phosphatases and other ectonucleotidases are membrane-bonded enzymes that play a significant role in the conversion of adenosine triphosphate (ATP) to adenosine [17]. The overproduction of this adenosine from ATP can trigger the progression of a malignant cancer in the form of multiple myeloma, breast cancer, myeloid leukemia, and liposarcomas [18,19,20,21,22]. In fact, an undesired formation of adenosine favors the stimulation of cancer and tumor growth through angiogenesis [23,24]. Towards this end, tissue-nonspecific alkaline phosphatase (TNAP) plays a major role in purinergic signaling in various cancers. Under normal physiological conditions, the body produces 20–200 nM of adenosine, while in the case of hypoxic conditions it increases up to approximately 10 μM. Therefore, the conversion of extracellular ATP into adenosine with the consequent stimulation of the adenosine subtype A2A and A2B receptors expressively contributes to the miserable diagnoses in numerous malignancies, such as triple-negative breast cancers, multiple myeloma, acute myeloid leukemia, and liposarcomas [22,25,26]. With this concept in mind, the development of novel and potent AP inhibitors that target adenosine signaling may be an effective way to resolve cancer-related complexities.

APs are also involved in the dephosphorylation of the proteins that play a critical role in the regulation of apoptosis [27]. Living organisms maintain the normal physiological functions of their bodies through apoptosis. The dysregulation of extracellular and intracellular factors has a prime role in the induction of apoptosis. The dysfunction of mitochondrial membranes, fragmentation of DNA, and activation of caspases are important biomarkers of apoptosis, which are initially triggered by reactive oxygen species. Reactive oxygen species (ROS) are considered to be involved in triggering, facilitating, and accomplishing apoptosis [28].

To combat cancer-related lethal diseases, several targeted therapies have been accepted for practice in the previous two decades for the treatment of various malignancies. These cancer treatments involve angiogenesis inhibitors, apoptosis inducers, gene expression modulators, immunotherapies, toxin delivery, signal transduction inhibitors, and hormone therapies [29]. Among these treatments, chemotherapy alone or in combination with other treatments is the best and safest option in treating various kinds of cancers [30,31]. The existing medications fail to distinguish between cancerous and normal cells due to the high toxicity profile of the currently available drugs [32]. With this reason in mind, there is a growing interest to focus on the rapid and cost-effective identification of new therapeutic targets, in which drugs only interact with the selected targets; this approach may found comparatively safe and effective approaches to treat various forms of cancer. In this respect, multi-component reactions are the best tool to address this challenge well. Multi-component reactions (MCRs) have gained much attention in the field of medicinal chemistry due to their wide range of applicability for the synthesis of complex molecular frameworks. The key feature of MCRs is that various types of bond breaking and making occur in a single step to offer a desired biologically active heterocyclic scaffold [33,34]. Currently, six-member heterocyclic chemistry has gained much attention due to its diverse biological applications [35]. Pyridine and dihydropyridines are reflected as vital drug candidates for the development of new therapeutic agents because these six-membered heterocycles with unique moieties are already known to present a range of biological activities and are found in most FDA-approved drugs [36,37]. Some FDA-approved drugs containing pyridine and dihydropyridine nuclei are vismodegib, [38] apatinib [39], clonixin [40], enasidenib [41], nevirapine [42], delafloxacine [43], nilvadipine [44], nimopidine [45], and nifedipine [46], as listed in Figure 1.

It was envisioned that the incorporation of various diverse structural features such as imidazole, sulfanilamide, and naphthyl moieties onto pyridine and dihydropyridine frameworks might enhance their anticancer potential. Therefore, as a part of our ongoing research on the synthesis and investigation of the biological activities of synthesized compounds, we synthesized structurally diverse analogs of pyridines and dihydropyridines and characterized them through spectroscopic techniques. Furthermore, the investigation of the in silico, in vitro enzyme inhibition potential and structural activity relationship (SAR) against nonspecific alkaline phosphatase (*h*-TNAP) supports each other. An MTT assay was used to investigate the neoplastic effect of the synthesized compounds through an initial screening on HeLa and MCF-7 cancer cell lines. The most promising analog **4d** was selected to explore its participation in the induction of apoptosis in cancerous cells. A high-resolution fluorescence microscope was used to ensure the viability and staining agents such as propidium iodide (PI) and 4′,6-diamidino-2-phenylindole (DAPI) were used to ensure the apoptosis in cells. Apoptosis induced through various external and internal factors such as DNA damage and dysfunction of mitochondria was further confirmed through the production of reactive oxygen species (ROS) by using a high-resolution fluorescence microscope imaging technique. Furthermore, the most active compound **4d** halted the cell cycle at the G_0_/G_1_ phase by inhibiting CDK4/6 and pRb proteins,; subsequently, minimum cells entered to S-phase. The binding energy and mode of DNA interactions with the compound **4d** were preliminarily confirmed through a molecular docking study and were further supported by fluorescence binding studies.

## 2. Results and Discussion

### 2.1. Rational Design Strategy

The contribution of 1,4-dihydropyridine- and pyridine-based scaffolds have been considerably recognized in the development of various drugs. In fact, these privileged scaffolds are part of many FDA-approved drugs for the treatment of various lethal diseases (Figure 1). Based on the wide range of the biological significance of dihydropyridine and pyridine moieties and their subsequent contribution towards the development of existing drugs, we were motivated to incorporate biologically active scaffolds in designing small organic molecules for the inhibition of specific enzymes and proteins. Therefore, a library of organic molecules has been designed around a specific scaffold, which allows us to systematically understand the biological potential of the scaffold and to further investigate biological activity space against target enzyme and proteins. In a small organic molecule, the tuning of substituents around biologically active scaffolds may change the functions of target proteins by inhibiting their normal functions. Structural modification through the presence of diverse functionalities on dihydropyridine and pyridine rings with variable chain lengths may improve the desired inhibition potential. In the current study, we decided to modify the structural framework of dihydropyridines and pyridines by introducing a range of moieties to accomplish interesting small heterocycles with a high inhibitory activity of alkaline phosphatases and cancer cell lines (HeLa and MCF-7). We have learned from our previous studies on the development of anticancer agents [47,48,49,50,51,52,53] that the potential of inhibitory activity can be enhanced through the incorporation of bulky groups with various spacers. Furthermore, pro-apoptotic behavior and cell cycle arrests were performed to assess the anticancer potential of the synthesized compounds. The present study is very critical in designing dihydropyridine and pyridine rings with suitable moieties that may interact with multi-targets to obtain an interesting anticancer potential. Systematic variations of the substitution pattern around the active scaffold could allow us to design and synthesize improved anticancer agents.

### 2.2. Chemistry

Pyridine and dihydropyridine scaffolds are present in most FDA-approved drugs due to their better compatibility with biological systems. There is a dire need to explore new derivatives and investigate their biological potential against various targeted enzymes in the search for novel successes for future drug development. In this regard, novel dihydropyridine (**4a**–**4f**) and pyridine derivatives (**4g**–**4k**) were developed with diverse functionality and showed strong interaction with the target enzymatic system, confirmed through in silico and in vitro studies. The multi-component reaction strategy was adopted to obtain the desired product with excellent yield from easily assessable, low-cost reagents. In the preliminary reaction, malononitrile **2a** was treated with different suitable aldehydes (**3a**–**3k**), particularly those containing electron-donating moieties, to develop an intermediate following the Knoevenagel reaction conditions, which further cyclized to a six-membered ring upon the addition of the appropriate amines (**1a**–**1d**). The general synthetic routes adopted for the synthesis of dihydropyridine (**4a**–**4f**) and pyridine derivatives (**4g**–**4k**) are summarized in Figure 1 and Figure 2. The basicity of the amines mainly decides the ring closer. The desired dihydropyridine (**4a**–**4f**) and pyridine analogs (**4g**–**4k**) were synthesized in good yields when the above-mentioned multi-component reaction was carried out at room temperature (25 °C) in the presence of a catalytic amount (20 mol%) of base (4-(dimethylamino) pyridine, DMAP) and while using methanol as solvent. All the synthesized compounds were purified through the column chromatography technique, and the purity of the compounds was investigated through the HPLC technique, as provided in the Appendix A.

Next, the scope of the reaction was extended by using different aliphatic amines under identical conditions to explore the effect of the nucleophilicity and basicity of the amine in the ring-closure reaction. Surprisingly, 1,4-dihydropyridine was produced instead of pyridine when 3-(1*H*-imidazol-1-yl)propan-1-amine **1c** and naphthaldehyde were utilized, which shows that naphthaldehyde resists the formation of pyridine. It is evidently clear from these results that the chemoselectivity relies not only on the basicity and nucleophilicity of amines as claimed in the previous reports [44], but also depends on the structure and nature of the used aldehyde.

The structures of the synthesized compounds were confirmed through NMR spectroscopy and mass spectrometry techniques. In the case of the dihydropyridines (**4a**–**4f**), the ^1^H NMR showed a characteristic singlet peak in the range of δ 3.96–4.52 ppm that corresponds to the methine (C–H) proton. Furthermore, the appearance of an intense singlet ranging from δ 6.15–6.47 ppm was attributed to four N–H protons, which validates the cyclization of the ring. The characteristic set of peaks between δ 6.37 and 7.95 ppm are an indication of aromatic protons. Additionally, the ^13^C and DEPT-135 NMR spectral data also confirmed the successful synthesis of the designed compounds. In the ^13^C NMR spectra, the carbon peaks in the range of δ 155.2–160.4, 50.7–60.1, and 40.5–50.9 ppm are correlated to the 2,6-C, C-(3,5), and C-4 of the dihydropyridines, respectively. The spectral peaks in the range of δ 45.2–50.7 ppm in DEPT-135 NMR are clear evidence of the methylene protons of the dihydropyridines. In addition to this, the ^1^H NMR spectral peaks at δ 3.98–4.56 ppm were absent in the ^1^H NMR spectra of the pyridine-based compounds (**4g**–**4i**). In the ^1^H NMR spectra of the pyridines, two broad signals appeared in the spectral range of δ 6.35–7.93 ppm and are validated to the NH and NH_2_ protons of the pyridines. Furthermore, the methylene protons attached to the nitrogen atom of the chain are represented by the appearance of doublet of triplets (dt) at δ 3.95 ppm. The triplet (t) at 2.95 ppm indicates the presence of a methylene group directly connected to the aromatic ring. The structure of the synthesized compounds was further confirmed through the NMR spectroscopy of ^13^C and DEPT-135. The characteristic carbon peaks at δ 161.2–159.1 and 80.4–79.0 ppm in the ^13^C NMR spectra of the pyridines further justified the ring closer. Two methylene proton peaks that appeared at δ 42.31–34.57 ppm in the DEPT-135 NMR confirmed the successful synthesis of pyridine scaffolds. In the ^1^H NMR spectra of the imidazole-substituted compounds (**4j**–**4k**), broad singlets at 6.80 and 7.46 ppm show the NH and NH_2_ protons. The three singlets in the range of 6.91–7.82 ppm in the aromatic ring also confirm the presence of the imidazole moiety in the parent structure. In the ^13^C NMR spectra of the imidazole-substituted compounds (**4j**–**4k**), carbon peaks shown at 170.0–160.4 ppm justify the ring formation. In the DEPT-135 NMR data, signals at 40.9–60.1 ppm correspond to the aliphatic region containing three CH_2_ groups.

### 2.3. Biological Studies

#### 2.3.1. Homology Modeling and Molecular Docking Studies of Most Potent Analogs against *h*-TNAP

We studied the most achievable binding interactions of the most biologically potent compounds (newly synthesized) within the active site of *h*-TNAP using the Molecular Operating Environment (MOE) software [54]. The amino acid residues which were involved in the bonding and nonbonding interactions with ligands **4d**, **4e**–**4g**, and **4j** include Arg167, Ser93, Asp92, His154, His321, His324, and His437, which played a significant role in ligand–protein binding [8,50]. The co-crystallized protein structure was not available, therefore a previously reported homology model of a protein (APs enzyme) was used for this object [54]. The sole purpose was to analyze the binding pattern of the newly synthesized compounds (**4d**, **4e**–**4g**, and **4j**) in the binding site of the *h*-TNAP enzyme. Protein was prepared for interaction using MOE and we selected the best conformations achieved from the interacting moieties. The analysis of the binding orientations and interaction plots was carried out by using the Discovery Studio Visualizer software.

The binding interactions of the levamisole within the active site of the target enzymes (APs) were studied as a reference using the Molecular Operating Environment (MOE) software [54]. The amino acid residues involved in the attractive charge included Arg167, Asp92, and Glu108. The π–π interaction was observed between the phenyl ring and His154, while the Zn ion interaction was also found to be the same as previously reported [54]. Results obtained through the in vitro analysis revealed that compound **4d** showed high potency against *h*-TNAP enzyme. The strong interaction between compound **4d** and the amino acid residue of an enzyme (h-TNAP) was further justified through an in silico approach. Strong hydrogen bonding (H-bonding) was observed between the amino groups of compound **4d** and amino acid residues Arg119 and Ser93, as shown in Figure 2. Moreover, the naphthyl group of compound **4d** interacted with His324 through the π–π interaction. The naphthyl group of compound **4d** also developed a strong interaction with His321 and His437. Similarly, the Zn438 of enzyme *h*-TNAP established a stable interaction with the naphthyl and amino group of compound **4d** [55]. Docking studies provided a detailed and deep throughput of the interaction of the compound and active pockets of the enzyme. Next, a *p*-methoxy substituent of compound **4e** made a strong hydrogen bonding interaction with the amino acid Arg119.

The carbon-hydrogen interaction was observed between the methoxy group of compound **4e** and amino acid residue Glu108. Similarly, the dihydropyridine core of compound **4e** interacted with His323 through a π–π interaction. The terminal phenyl ring of compound **4e** was able to interact through π–π, π–cation, and π–anion associations with different amino acids, i.e., Asp320, Arg167, His324, and Zn483, respectively. The *m*-hydroxyl unit of compound **4f** interacted through a strong hydrogen bonding with amino acid Arg151, whereas a carbon-hydrogen interaction was observed between the methyl unit of compound **4e** and amino acid residue Asp277. The pyridine ring of compound **4f** developed a π–π interaction with amino acid residue His434. Moreover, various interactions were observed between the aryl ring of compound **4f** and amino acids residue Asp320, His324, and Zn483. Moreover, the peripheral benzyl ring of compound **4f** made π–π and π–cation interactions with amino acid His 324, Asp 320, and Zn 483.

Compound **4g** was found to be very potent, with an IC_50_ value of 0.74 ± 0.034 µM. The molecular docking studies showed that molecules had good interactions with the key amino acids. There was a strong hydrogen bonding of compound **4g** with the Arg167 and His437 amino acids, while a carbon-hydrogen interaction was observed with His324 and Thr436. Compound **4g** developed a π–cation interaction with His154 and His321 that has a close resemblance to reported data [54]. Compound **4j** was active, with an activity (IC_50_) of 5.34 ± 0.16 µM. In molecular docking, hydrogen bonding was observed between the cyanide group at position three (3) with Arg151 and nitrogen of the dihydropyridine ring with amino acid Arg167, while the aryl ring established a π–π interaction with His324 and His434. In compound **4j**, a carbon-hydrogen (C-H) interaction was examined between its cyano group and amino acid His154, and a π–anion interaction was observed between the imidazole ring and Asp320 and zinc ion, as shown in Figure 2. The binding energies of the compounds (**4d**, **4e**, **4f**, **4g**, and **4j**) and levamisole were −5.4753, −3.0921, −4.1243, −6.2850, −5.4753, and −5.8749 kcal/mol respectively.

#### 2.3.2. Alkaline Phosphatase Inhibition Potential Studies and Structure–Activity Relationship (SAR)

Dihydropyridine and pyridine scaffolds are known as medicinally privileged nuclei and have anticancer activities and unique multifunctional properties. Designing new structures with which to control biological activity requires a systematic structural variation on the medicinally active scaffolds. With this in mind, various dihydropyridine and pyridine derivatives with remarkable therapeutic effects were designed and investigated for their inhibitory potency against human tissue nonspecific alkaline phosphatase (*h*-TNAP). Furthermore, the structure–activity relationship (SAR) is a very attractive tool for investigating the correlation of the enzyme inhibition potential on the structural feature of target heterocycles for new drug discovery and lead optimization. The structural nature of the basic core and the presence of suitable functional groups on the parent scaffold play a key role in enabling the interaction of the heterocycle scaffold with the target enzyme. This kind of efficient interaction of heterocycle scaffold with the target enzyme significantly affects the enzyme activity. More specifically, the entire structural unit and position of the functional groups considerably change the inhibitory potential of the target enzyme. As expected, most of the compounds have shown a remarkable inhibition potential against *h*-TNAP in comparison to a standard drug (levamisole). To understand the relationship of the structural variation of the synthesized compounds with the inhibitory activity of the *h*-TNAP enzyme, various analogs of dihydropyridine and pyridine were designed, synthesized, and further screened against the *h*-TNAP enzyme (Table 1). The general SAR was established based on the inhibitory activity of the *h*-TNAP, as revealed in Table 2. Compounds **4a**–**4k**, with a different nature of substituents, were initially tested against the *h*-TNAP enzyme (Table 1). The dihydropyridine-based compounds **4a** and **4b** showed the least activity against *h*-TNAP due to the presence of hydrophobic groups at the para-position of the phenyl ring that cannot develop an effective interaction with the amino acid residues present in the active pocket of the *h*-TNAP enzyme (Table 1). Likewise, the 1-naphthyl-substituted compound **4c** exhibited minimum inhibitory potency (26.19%) against *h*-TNAP. Surprisingly, compound **4d** which contains 2-naphthyl showed exceptional inhibitory activity with an IC_50_ value of 1.32 ± 0.26 µM against *h*-TNAP, which is about seventeen-fold more potent than the standard drug levamisole (Table 1). The assessment from the docking results is that the position and orientation of the naphthyl moieties play an effective role in the inhibition of enzymatic activity. The aromatic rings of the 2-naphthyl-substituted dihydropyridine **4d** displayed an excellent inhibitory activity due to π−π stacking and π−π T-stacking interactions with the amino acids in the active pocket of the enzyme. The extent of this π−π stacking interaction depends on the typical orientation of the respective naphthyl moiety. Similarly, the dihydropyridines **4e** and **4f** containing 2,3,4-trimethoxy and 2-methoxy-3-hydroxyl substituents on the phenyl ring displayed excellent inhibitory activities, with IC_50_ values of 8.8 ± 0.53 µM, 2.25 ± 0.16 µM and 3.21 ± 0.09 µM, respectively (Table 1). These excellent inhibitory activities of the dihydropyridine-based compounds **4e**–**4f** is attributed to the strong hydrogen bonding interactions of the polar groups with amino acid residue of the *h*-TNAP enzyme.

In the case of the sulfanilamide-based pyridine analogs **4g**–**4i**, the compound **4g** containing the *p*-ethoxy group presented an outstanding inhibitory potency (IC_50_ = 0.49 ± 0.025 µM), which is about thirty-fold higher than the inhibition potential of the referenced standard drug levamisole (IC_50_ = 22.65 ± 1.60 µM) (Table 1). On the contrary, pyridine-based compounds **4h** and **4i** showed the least inhibitory activity against the *h*-TNAP enzyme (Table 1). In these compounds, a significant loss in inhibitory potential was observed by replacing the polar ethoxy group with nonpolar thiomethyl and isopropyl groups. In fact, thiomethyl (-SCH_3_) and isopropyl groups could not develop favorable interactions with the active site of the enzyme due to a lack of hydrogen donor and hydrogen acceptor moieties. The nature of nonpolar substituents on the phenyl ring is responsible for badly affecting the inhibitory activity of the compounds due to the absence of hydrogen bonding type interaction, and their steric bulk on the phenyl ring further prevents the π–π stacking interaction of the phenyl group with aromatic group of amino acids in the active pocket of enzyme.

With regard to the further mapping of the inhibitory activity profile, more compounds were designed by replacing the sulfanilamide moiety with an imidazole moiety (Table 1). The replacement of the sulfanilamide moiety with an imidazole ring was initiated in the search to find more potent compounds that bind to the *h*-TNAP enzyme with improved affinity. With this intent, imidazole-based dihydropyridine and pyridine derivatives were synthesized and screened to further investigate their inhibitory activity toward the *h*-TNAP enzyme (Table 1). Amazingly, the imidazole-pyridine based compound **4j** with *p*-ethoxy substituent on the para-position of the phenyl ring was found to be a strong inhibitor, with an IC_50_ value of 1.22 ± 0.12 µM, about four-fold higher than levamisole. However, in this series, the compound **4k** having a *p*-isopropyl group was comparatively inactive against *h*-TNAP. The obtained results reveal that the ethoxy substituent at the para-position of the aromatic ring plays an important role in enhancing the inhibition potential against *h*-TNAP due to the strong hydrogen bonding with the amino acid residue of *h*-TNAP.

To sum up the structure–activity relationship based on docking studies and dose dependent response curve obtained from *h*-TNAP assay, most of the synthesized compounds manifested a higher inhibition potential than the standard drug levamisole (Figure 3). Inhibition potential of synthesized compounds is strongly dependent on the nature and position of substitutions over the aromatic ring attached to the dihydropyridine and pyridine basic nucleus.

#### 2.3.3. In Vitro MTT Cell Viability Assay

An MTT assay was used to determine the viability of the cells under examination in the desired medium. The mitochondrial dehydrogenase enzyme of the viable cells has a key role in the conversion of the yellow water-soluble substrate 3-(4,5-dimethylthiazol-2-y1)-2,5 diphenyl tetrazolium bromide (MTT) into a dark blue formazan product that is insoluble in water [56]. The anticancer activity of the novel synthetic derivatives (**4a**–**4k**) was investigated through an MTT assay, with small modifications following the already published protocols against cancer cell lines (HeLa and MCF-7), using cisplatin as a reference drug [57]. The cytotoxic effect of the tested compounds at 100 µM concentration was determined and represented as the percent growth inhibition compared with the untreated control cells. The first series comprised dihydropyridines analogous from **4a**–**4f** with various substitution patterns on the phenyl ring. In this series, the compounds **4a** and **4b** showed the least potency against both cancer cell lines (Hela and MCF-7) due to the hydrophobic group present at the para-position of the phenyl ring. The 2-Naphthyl-substituted compound **4d** showed the highest inhibitory potential against both cell lines (HeLa and MCF-7) with IC_50_ values of 53.47 ± 0.50 µM and 38.71 ± 2.31 µM, respectively (Table 2). However, compound **4c** with the 1-naphthyl unit remained inactive against the HeLa and MCF-7 cancer cell lines. It is evident from these results that the position and orientation of the naphthyl moiety plays a significant role in exhibiting the inhibitory potency against cancer cell lines. It is presumed that the 1-naphthyl containing compound **4c** was locked in a configuration that is not favorable for an interaction with amino acid residues. The compound **4e** was active against the HeLa cell lines with IC_50_ values of 69.47 ± 0.07 µM.

Similarly, the compound **4f** has 2-methoxy-3-hydroxy functionality. The compound **4f** was active against HeLa cell lines with IC_50_ values of 57.03 ± 7.1 µM. In the second series, the compound **4g** containing an ethoxy substituent further enhanced the activity against cancer cell lines. Additionally, the incorporation of sulfanilamide functionality in the pyridine framework (compounds **4h**–**4i**) caused a decrease in their activity against both cell lines. In the third series, the molecular hybridization of imidazole with the pyridine moiety enhanced the inhibition potential against HeLa cell lines. Among these, compound **4j** and **4k** displayed significant inhibition potential, with IC_50_ values of 57.01 ± 1.5 µM and 65.95 ± 1.97 µM, respectively (Table 2). The most active compound **4d** was tested on normal BHK-21 cells for an evaluation of its cytotoxic effect on rapidly growing non-cancerous cells, and an IC_50_ value of 71.07 ± 3.18 µM was calculated through an MTT assay.

#### 2.3.4. Observation of Cytotoxic Activity through PI and DAPI Staining

Change in the cellular morphology, initially characterized by cell shrinkage, chromatin condensation, and degradation of the nuclear and plasma membranes, was mostly observed in the programmed cell death [58]. The propidium iodide (PI) staining agent is a fluorescent molecule that penetrates the cancerous cells through ruptured membranes and binds with DNA. Similarly, the subsequent image offers red nuclei contrary to the dark background that indicates the counts of dead cells. On the other hand, viable cells have an intact cell membrane that resists the penetration of the staining agent (PI) [59,60]. Herein, the anticancer potential of the highly functionalized small dihydropyridine scaffold **4d** was noticed through the apoptosis induction pathway based on various mechanisms. The most active compound **4d** was assessed on behalf of its cytotoxic potential with IC_50_ and 2 × IC_50_ values of 28.3 µM and 56.6 µM, respectively, attained through a fluorescence microscopy that used propidium iodide (PI) as a staining agent. The detachment of cells and appearance of yellow intense fluorescence images on the dark background specify the distribution of PI into the dying cells in a dose-reliant mode.

The apoptotic feature, particularly the condensation of nuclear material, was also observed through the fluorescence microscopic evaluation of compound **4d**-treated HeLa cells using 4′,6-diamidino-2-phenylindole (DAPI) as a staining agent. Compound **4d**-treated cells showed cellular shrinkage and nuclear condensation in a dose-dependent manner. The staining agents PI and DAPI enter cancerous cells through ruptured permeable plasma membrane and interact with DNA. The fluorescence intensity enhances many folds upon binding with DNA. The extent of damage in the compound-treated cells was easily measured through an increase in the fluorescence emission intensity. Among the various analogs of pyridines and dihydropyridines, the most effective derivative **4d** tempted the highest apoptosis, as displayed by the DAPI staining agent. Compound-treated cells showed a bright cellular shrinkage and fragmented nuclei in a dose-dependent manner, as shown in Figure 4. Unprocessed cells presented no variation in their morphology [61].

The PI dye showed red fluorescence emissions having excitation and emission maxima in the range between 535 nm and 617 nm, respectively. PI-stained HeLa cell lines treated with IC_50_ and 2 × IC_50_ of compound **4d** under a fluorescence microscope displayed bright yellow emission of nuclei when compared with the untreated PI-stained control HeLa cell lines, which appeared as a non-fluorescent field. Similarly, the DAPI dye-stained HeLa cell lines showed a blue fluorescence emission having excitation and an emission maxima in the range between 350 nm and 460 nm, respectively, when it binds with DNA. However, the DAPI-stained HeLa cells exhibited a bright emission upon the treatment with IC_50_ and 2 × IC_50_ of compound **4d** (Figure 4). The cytotoxic activity of compound **4d** was also compared with cisplatin, which presented a clear comparison of the activities of the standard drug and designed compound. These results clearly validate the cytotoxic activity of the highly potent compound **4d** on cancer cell lines.

#### 2.3.5. Observation of ROS Production

The production of reactive oxygen species (ROS) has been an extensively researched topic in medicinal chemistry due to their involvement in numerous biochemical phases of health and disease [62]. In a biological system, redox-sensitive fluorogenic materials such as 2′,7′-dichlorodihydrofluorescein-diacetate (DCFH2-DA) are commonly used to investigate the production of ROS during oxidative stress. It easily penetrates through the cell membrane, where esterase enzyme hydrolyzes the acetate group to a more hydrophilic group, which further oxidizes in the cytoplasmic region to give rise to a highly fluorescent product [63]. In this study, the HeLa cells were used to investigate the production of ROS. The non-fluorescent material 2′,7′-dichlorodihydrofluorescein (DCFH2) converts into the fluorescent 2′,7′-dichlorofluorescein (DCF) under intracellular oxidative stress [64]. The rise in reactive oxygen species (ROS) after treatment with most active compound **4d** was assessed through a fluorescence microscopy using H2DCF-DA dye, as illustrated in Figure 5. The most potent compound **4d** showed a dose-dependent increase in the production of ROS in the compound-treated cells as compared with the control. When studied under a fluorescent microscope having a green filter, they had a wavelength of λ_ex_ = 495 nm, λ_em_ = 525 nm.

The HeLa cells treated with the IC_50_ and 2 × IC_50_ of compound **4d** displayed a bright green fluorescence emission of cells under a fluorescence microscope when compared with the untreated control HeLa cell lines, which appeared as a non-fluorescent field. These results evidently support the increased extent of ROS production when using the compound **4d**. The overexpression of ROS causes DNA damage and mitochondrial dysfunction that facilitates the release of cytochrome c, which further activates the caspases that lead to apoptosis. Collectively, these results demonstrate that the compound **4d** can enhance reactive oxygen species levels in a cancerous medium and may cause the death of cells through apoptosis.

#### 2.3.6. Cell Cycle Arrest

Based on our analysis, compound **4d** demonstrated the most potent activity against breast cancer cell lines (MCF-7). The sample was examined at final concentrations of IC_50_ and 2 × IC_50_ using the flow cytometry technique. As shown in Figure 6, compound **4d** revealed the extent of activity that DNA content (breast cancer cell DNA) was distributed in the different phases G_0_/G1, S, and G2/M after being stained with the fluorescent propidium iodide (PI) dye. A cell cycle arrest was noticed in the G_0_/G_1_ phase after 24 h of treatment with both concentrations (IC_50_ and 2 × IC_50_) of compound **4d**. An arrest in the cell cycle via significant inhibition in the cell proliferation was observed in the G_0_/G_1_ phase. The progression of the cell cycle is started by a cascade of intracellular signaling pathways to drive progression from the G_0_ phase to the G_1_ phase. The mammalian cell cycle involves an overexpression of D-type cyclins during the progression from the G_0_/G_1_ to the S-phase and eventually contributes to the development of cancer [65]. Progression is controlled by D-type cyclins after binding with cyclin-dependent kinases (CDKs) and negatively controlled by binding to CDK inhibitors [66,67]. CDK4/6 is involved in the G_1_/S transition of the cell cycle and has an important role in breast cancer. Retinoblastoma tumor suppressor protein (pRb) is deactivated through a CDK4/6-mediated phosphorylation reaction. In fact, cyclin D1/CDK4/6 is a mandatory controller of pRb phosphorylation. This phosphorylation deactivates the pRb that leads to cell proliferation. The most active compound **4d** halted the cell cycle at the G_0_/G_1_ phase by inhibiting CDK4/6 and the subsequent activation of pRb proteins. It is believed that the inhibition of CDK4/6 and subsequent activation of pRb proteins significantly reduces cells entry to the S–phase [66].

#### 2.3.7. In Silico Studies of Compound **4d** with DNA

The docked structure as presented in Figure 7 shows that compound **4d** binds with the minor groove of DNA. Both amine groups of compound **4d** are involved in the formation of conventional hydrogen bonds with the side chain residues of DNA. The first amine group of compound **4d** forms a hydrogen bond (2.36 Å) with the oxygen atom at the third position of sugar of cytosine 21 (Chain B). The second hydrogen bond interaction (2.37 Å) is formed between the second amine group of ligand **4d** and phosphate oxygen of side chain thymine 7 (Chain A). Furthermore, the complex is stabilized (−7.4 kcal·mol^−1^) by the hydrophobic interactions formed between the aromatic rings of compound **4d** and side chain residues, i.e., alanine 6A, thymine 7A, thymine 8A, thymine 20B, and cytosine 21B; These include the formation of π–anion and π–σ interactions between the naphthyl group of compound **4d** and phosphate and methylene units of thymine 8A, respectively. The terminal phenyl group is involved in the formation of a π–anion interaction with the phosphate group of cytosine 21B.

#### 2.3.8. Compound **4d**-DNA Interaction through Fluorescent Binding

The fluorescence emission studies of compound **4d** (10 µM) were accomplished in DMF/H_2_O (1:9, *v*/*v*) before and after the continual addition of DNA (0–50 µg/mL) to probe out the compound-DNA interaction. In the preliminary study, compound **4d** was found to be highly emissive at 410 nm. As evident in Figure 8, the fluorescence emission of compound **4d** was attenuated by ~55% (calculated through I_0_/I) upon the maximum addition of 50 µg/mL of DNA. The substantial fluorescence quenching behavior of **4d** upon the addition of DNA is perhaps attributed to the efficient binding of the compound to the minor groove of DNA [68]. Different electrostatic interactions such as hydrogen bonding (H-bonding) or van der Waals are commonly involved in the potential binding of smaller organic molecules with minor grooves of DNA. Since the adenine-thymine (A-T) groove regions are narrow compared with the guanine-cytosine (G-C) grooves, compound **4d** is liable to interact with AT regions. The inability of compound **4d** to bind with G-C regions is probably due to the presence of a sterically hindered amino group in the G-C grooves. Next, the excellent binding efficiency of compound **4d** with DNA was well elucidated through the Stern–Volmer (SV) plot, which was produced between the increased fluorescence quenching efficiency and increasing concentration of DNA solution. The SV plot shown in Figure 8 displays a linear behavior against the increasing concentrations of DNA solution (0–50 µg/mL). The linear region in the SV plot provides the Stern–Volmer constant (K_sv_) value, which was estimated at 3.41 × 10^4^ M^−1^, showing the excellent binding capacity of compound **4d** with DNA.

## 3. Materials and Methods

### 3.1. Instruments and Reagents

Reagent grade solvents were used for purification purposes. Silica gel with 70–230 mesh size and pure solvents were used for the column chromatography. ^1^H, ^13^C, and DEPT-135 NMR spectra were recorded on a 400 MHz spectrometer using DMSO as the solvent and TMS as the internal standard. The chemical shifts, multiplicity, and coupling constants were calculated from data obtained through the FID file. The docking study was performed through the MOE software. A basic buffer with a pH of 9.5 was used for the assay of the *h*-TNAP enzyme. The IC_50_ and percent inhibition were calculated by using GraphPad software. We seeded 1 × 10^4^ cells/well in a sterile, 96-well culture microtiter plate and incubated them in a 5% CO_2_ incubator at 37 °C for 24 h. The optical density was measured through a microplate reader at 570 nm. 4′,6-diamidino-2-phenylindole (DAPI) and propidium iodide (PI) dyes were used as staining agents. The fluorescent images were captured through a high-resolution fluorescence microscope (Nikon ECLIPSE Ni–U). Cells were grown at a density of 2 × 10^5^ cells/well/mL on round coverslips in a 24-well plate. Dichlorofluorescin diacetate (H2DCF-DA) dye was used for the detection of reactive oxygen species (ROS). Samples were made to run using the BD Accuri C6 flow cytometer, and 10,000 events were taken. The obtained results were analyzed by using the BD Accuri^TM^ C6 software and GraphPad Prism 5.01. Herring sperm DNA was used for the binding study.

### 3.2. General Procedure for the Preparation of Dihydropyridines and Pyridines

The mixture of appropriate aldehyde (5.0 mmol), malononitrile (10.0 mmol), and 20 mol% dimethyl aminopyridine (DMAP) was dissolved in HPLC grade methanol (30.0 mL). The mixture was stirred for 1 h at room temperature (25 °C). After a few minutes, the precipitates appeared that were an indication of the formation of reactive intermediates following the Knoevenagel condensation reaction. An appropriate amine (10.0 mmol) was added to the reaction mixture, which resulted in the sudden disappearance of solid precipitates. The reaction mixture was left for vigorous stirring. After 10 h of continuous stirring, the reaction mixture solidified, which indicated the completion of the reaction. Precipitates were filtered through a Buckner funnel followed by washing with an excess of ethanol. Further purification was carried out through the column chromatography technique using petroleum ether and ethyl acetate (50:50) as an eluent to obtain a pure product.

### 3.3. Spectral Data of Compounds *(**4a**–**4l**)*

#### 3.3.1. 2,6-Diamino-1-benzyl-4-(4-(methylthio)phenyl)-1,4-dihydropyridine-3,5-dicarbonitrile (**4a**)

^1^H NMR (400 MHz, DMSO-*d*_6_) δ: 7.34–7.32 (m, 3H, Ar-H), 7.20 (dd, 2H, *J =* 7.2, 3.7 Hz, Ar-H), 7.06 (d, 2H, *J =* 8.3 Hz Ar-H), 6.81 (d, 2H, *J =* 8.2 Hz Ar-H), 6.26 (s, 4H, 2 × NH_2_), 4.96 (s, 2H, CH_2_), 3.96 (s, 1H, CH of DHP ring), 2.44 (s, 3H, CH_3_). ^13^C NMR (100 MHz, DMSO-*d*_6_) δ: 152.4 (2 × C), 142.2 (1 × C), 136.7 (1 × C), 136.1 (1 × C), 128.6 (2 × C), 127.8 (2 × C), 127.4 (2 × C), 126.1 (1 × C), 121.5 (2 × C), 61.3 (2 × C), 46.9 (1 × C), 38.9 (1 × C), 15.0 (1 × C). LRMS [ESI]^−^: *m*/*z* calcd for C_21_H_19_N_5_S [M − H]^−^ 372.1; found 372.2.

#### 3.3.2. 2,6-Diamino-1-benzyl-4-(4-isopropylphenyl)-1,4-dihydropyridine-3,5-dicarbonitrile (**4b**)

^1^H NMR (400 MHz, DMSO-*d*_6_) δ: 7.32–7.30 (m, 3H, Ar-H), 7.20 (dd, 2H, *J**=* 7.4, 2.0 Hz, Ar-H), 7.03 (d, 2H, *J*
*=* 8.1 Hz, Ar-H), 6.81 (d, 2H, *J*
*=* 8.1 Hz, Ar-H), 6.23 (s, 4H, 2 × NH_2_), 4.96 (s, 2H, CH_2_), 3.94 (s, 1H, CH of DHP ring), 2.82 (sep, *J =* 6.9 Hz, 1H, CH of isopropyl), 1.18 (d, *J =* 6.9 Hz, 6H, 2 × CH_3_). ^13^C NMR (100 MHz, DMSO-*d*_6_) δ: 152.4 (2 × C), 146.7 (1 × C), 142.8 (1 × C), 136.8 (1 × C), 128.5 (2 × C), 127.8 (2 × C), 127.7 (1 × C), 126.7 (2 × C), 126.2 (2 × C), 121.7 (2 × C), 61.5 (2 × C), 46.8 (2 × C), 33.1 (1 × C), 24.0 (2 × C); LRMS [ESI]^+^: *m*/*z* calcd for C_23_H_23_N_5_ [M − H]^−^ 368.2; found 368.3.

#### 3.3.3. 2,6-Diamino-1-benzyl-4-(naphthalen-1-yl)-1,4-dihydropyridine-3,5-dicarbonitrile (**4c**)

^1^H NMR (400 MHz, DMSO-*d*_6_) δ: 7.85 (d, *J =* 8.8 Hz, 1H, Ar-H), 7.74 (d, *J =* 7.3 Hz, 1H, Ar-H), 7.71 (d, *J =* 8.5 Hz, 1H, Ar-H), 7.52–7.45 (m, 3H, Ar-H), 7.33–7.24 (m, 5H, Ar-H), 7.02 (dd, *J =* 8.4, 1.5 Hz, 1H, Ar-H), 6.32 (s, 4H, 2 × NH_2_), 5.00 (s, 2H, CH_2_), 4.19 (s, 1H, CH); ^13^C NMR (100 MHz, DMSO-*d*_6_) δ: 152.5 (2 × C), 142.8 (1 × C), 136.8 (1 × C), 132.8 (1 × C), 132.2 (1 × C), 128.6 (2 × C), 128.2 (1 × C), 127.8 (2 × C), 128.0 (1 × C), 127.8 (1 × C), 127.5 (1 × C), 126.2 (1 × C), 125.7 (1 × C), 125.6 (1 × C), 124.7 (1 × C), 122.0 (2 × C), 61.1 (2 × C), 46.8 (1 × C); LRMS [ESI]^+^: *m*/*z* calcd for C_24_H_19_N_5_ [M − H]^−^ 376.2; found 376.2.

#### 3.3.4. 2,6-Diamino-1-benzyl-4-(naphthalen-2-yl)-1,4-dihydropyridine-3,5-dicarbonitrile (**4d**)

^1^H NMR (400 MHz, DMSO-*d*_6_) δ: 7.85 (d, *J =* 8.9 Hz, 1H, Ar-H), 7.74 (d, *J =* 8.9 Hz, 1H, Ar-H), 7.71 (d, *J =* 8.6 Hz, 1H, Ar-H), 7.52–7.45 (m, 3H, Ar-H), 7.34–7.25 (m, 5H, Ar-H), 7.02 (dd, *J =* 8.4, 1.7 Hz, 1H, Ar-H), 6.32 (s, 4H, 2 × NH_2_), 5.01 (s, 2H, CH_2_), 4.20 (s, 1H, CH); ^13^C NMR (100 MHz, DMSO-*d*_6_) δ: 152.5 (2 × C), 142.8 (1 × C), 136.8 (1 × C), 132.8 (1 × C), 132.2 (1 × C), 128.6 (2 × C), 128.2 (1 × C), 127.8 (2 × C), 128.0 (1 × C), 127.8 (1 × C), 127.5 (1 × C), 126.2 (1 × C), 125.7 (1 × C), 125.6 (1 × C), 124.7 (1 × C), 121.6 (2 × C), 61.1 (2 × C), 46.8 (1 × C); LRMS [ESI]^+^: *m*/*z* calcd for C_24_H_19_N_5_ [M − H]^−^ 376.2; found 376.3.

#### 3.3.5. 2,6-Diamino-1-benzyl-4-(2,3,4-trimethoxyphenyl)-1,4-dihydropyridine-3,5-dicarbonitrile (**4e**)

^1^H NMR (400 MHz, DMSO-*d*_6_) δ: 7.39–7.35 (m, 2H, Ar-H), 7.30–7.25 (m, 2H, Ar-H), 7.21 (d, *J =* 7.3 Hz, 2H, Ar-H), 7.03 (d, *J =* 8.6 Hz, 1H, Ar-H), 6.92 (d, *J =* 8.7 Hz, 1H, Ar-H), 6.53 (s, 1H, -CH), 5.55 (br. s, 1H, CH_2_), 5.49 (br. s, 1H, -CH_2_), 3.87 (s, 3H, -OCH_3_), 3.82 (s, 3H, -OCH_3_), 3.79 (s, 3H, -OCH_3_); ^13^C NMR (100 MHz, DMSO-*d*_6_) δ: 154.8 (2 × C), 154.6 (1 × C), 150.2 (1 × C), 141.5 (1 × C), 134.9 (2 × C), 128.9 (2 × C), 128.5 (2 × C), 127.2 (2 × C), 126.4 (1 × C), 123.8 (2 × C), 121.8 (2 × C), 115.5 (2 × C), 111.7 (2 × C), 107.8 (2 × C), 61.3 (2 × C), 60.7 (2 × C), 56.00 (1 × C), 45.5 (2 × C), 29.8 (1 × C); LRMS [ESI]^+^: *m*/*z* calcd for C_23_H_23_N_5_O_3_ [M − H]^−^ 416.2; found 416.3.

#### 3.3.6. 2,6-Diamino-1-benzyl-4-(3-hydroxy-2-methoxyphenyl)-1,4-dihydropyridine-3,5-dicarbonitrile (**4f**)

^1^H NMR (400 MHz, DMSO-*d*_6_) δ: 8.47 (s, 1H, Ar-OH), 7.39–7.28 (m, 5H, Ar-H), 6.69 (dd, *J* = 8.0, 1.1 Hz, 1H, Ar-H), 6.41 (t, *J* = 7.9 Hz, 1H, Ar-H), 6.10 (br. s, 4H, 2 × NH_2_), 6.07 (d, *J* = 7.8 Hz, 1H, Ar-H), 4.97 (s, 2H, -CH_2_), 4.49 (s, 1H, CH), 3.75 (s, 3H, -OCH_3_); ^13^C NMR (100 MHz, DMSO-*d*_6_) δ: 152.6 (2 × C), 146.9 (1 × C), 143.1 (1 × C), 136.8 (1 × C), 131.7 (1 × C), 128.5 (2 × C), 128.1 (2 × C), 127.8 (1 × C), 121.5 (1 × C), 120.3 (1 × C), 118.5 (2 × C), 106.6 (1 × C), 61.6 (2 × C), 55.8 (1 × C), 46.7 (1 × C), 31.2 (1 × C); DEPT-135 NMR (100 MHz, DMSO-*d*_6_) δ: 128.5, 128.1, 127.8, 120.3, 118.5, 109.6, 55.8, 46.7, 31.2; LRMS [ESI]^+^: *m*/*z* calcd for (C_21_H_19_N_5_O_2_ [M + H]^+^ 372.2; found 372.3.

#### 3.3.7. 4-(2-((6-Amino-3,5-dicyano-4-(4-ethoxyphenyl)pyridine-2yl)amino)ethyl)benzene sulfonamide (**4g**)

^1^H NMR (400 MHz, DMSO-*d*_6_) δ: 7.76 (d, *J =* 8.2 Hz, 2H, Ar-H), 7.54 (t, *J =* 5.6 Hz, 1H, Ar-H), 7.48 (d, *J =* 8.3 Hz, 2H, Ar-H), 7.43–7.40 (m, 3H, Ar-H & -NH), 7.30 (s, 2H, Ar-H), 7.05 (d, *J =* 8.7 Hz, 2H, Ar-H), 4.10 (q, *J =* 6.9 Hz, 2H, CH_2_), 3.61–3.56 (m, 2H, CH_2_), 2.95 (t, *J =* 8.0 Hz, 2H, CH_2_), 1.36 (t, *J =* 6.9 Hz, 3H); ^13^C NMR (100 MHz, DMSO-*d*_6_) δ: 161.2 (1 × C), 159.9 (1 × C), 159.3 (1 × C), 159.2 (1 × C), 143.8 (1 × C), 142.2 (1 × C), 130.1 (2 × C), 129.4 (2 × C), 128.9 (1 × C), 127.0 (2 × C), 125.8 (1 × C), 116.9 (1 × C), 116.7 (1 × C), 114.4 (2 × C), 80.5 (1 × C), 79.1 (1 × C), 63.4 (1 × C), 42.3 (1 × C), 34.6 (1 × C), 14.7 (1 × C); LRMS [ESI]^+^: *m*/*z* calcd for C_23_H_22_N_6_O_3_S [M − H]^−^ 461.2; found 461.3.

#### 3.3.8. 4-(2-((6-Amino-3,5-dicyano-4-(4-(methylthio)phenyl)pyridin-2-yl)amino)ethyl)benzene sulfonamide (**4h**)

^1^H NMR (400 MHz, DMSO-*d*_6_) δ: 7.76 (d, *J =* 8.2 Hz, 2H, Ar-H), 7.59 (t, *J =* 5.5 Hz, 1H, Ar-H), 7.48 (d, *J =* 8.2 Hz, 3H, Ar-H), 7.43–7.38 (m, 5H, Ar-H), 7.30 (s, 2H, SO_2_NH_2_), 3.61–3.56 (m, 2H, CH_2_), 2.95 (t, *J =* 8.0 Hz, 2H, -CH_2_), 2.54 (s, 3H, -SCH_3_); ^13^C NMR (100 MHz, DMSO-*d*_6_) δ: 161.2 (1 × C), 159.1 (1 × C), 143.7 (1 × C), 142.2 (1 × C), 141.1 (1 × C), 131.1 (1 × C), 129.4 (2 × C), 129.0 (2 × C), 125.8 (2 × C), 125.3 (2 × C), 116.8 (1 × C), 116.6 (1 × C), 80.4 (1 × C), 79.0 (1 × C), 42.3 (1 × C), 34.6 (1× C), 14.2 (1 × C); DEPT-135 NMR (100 MHz, DMSO-*d*_6_) δ: 129.4, 129.0, 125.8, 125.3, 42.3, 40.1, 39.9, 39.6, 34.6, 14.2; LRMS [ESI]^+^: *m*/*z* calcd for C_22_H_20_N_6_O_2_S_2_ [M − H]^−^ 463.2; found 463.3.

#### 3.3.9. 4-(2-(6-Amino-3,5-dicyano-4-(4-isopropylphenyl)pyridin-2-yl)amino)ethyl)benzene sulfonamide (**4i**)

^1^H NMR (400 MHz, DMSO-*d*_6_) δ: 8.32 (s, 2H), 7.78 (d, *J* = 8.3 Hz, 2H, ArH), 7.56 (d, 2H, *J* = 8.3 Hz), 7.41 (br. s, 4H, ArH), 7.31 (s, 2H, ArH), 4.37 (br. s, 2H, -CH_2_), 3.01–2.94 (m, 3H, CH & CH_2_), 1.99 (s, 1H), 1.25 (d, 6H, *J* = 6.9 Hz, 2xCH_3_); LRMS [ESI]^+^: *m*/*z* calcd for C_24_H_24_N_6_O_2_S [M − H]^−^ 459.2; found 459.3.

#### 3.3.10. 2-((3-(1H-Imidazol-1-yl)propyl)amino)-6-amino-4-(4-ethoxyphenyl)pyridine-3,5-dicarbonitrile (**4j**)

^1^H NMR (400 MHz, DMSO-*d*_6_) δ: 7.65 (s, 1H, imidazole), 7.39 (d, *J =* 4.7 Hz, 2H, Ar-H), 7.19 (s, 1H, imidazole), 7.04 (d, *J =* 8.8 Hz, 2H, Ar-H), 6.89 (s, 1H, imidazole), 4.17 (br. s, NH), 4.12–4.04 (m, 4H, 2 × CH_2_), 2.05 (q, *J =* 7.5 Hz, 2H, CH_2_), 1.35 (t, *J =* 7.0 Hz, 3H, CH_2_); ^13^C NMR (100 MHz, DMSO-*d*_6_) δ: 160.0 (1 × C), 156.2 (1 × C), 137.2 (1 × C), 129.8 (2 × C), 128.5 (2 × C), 126.9 (1 × C), 119.3 (1 × C), 117.4 (1 × C), 114.3 (1 × C), 63.4 (1 × C), 43.7 (1 × C), 40.9 (1 × C), 27.2 (1 × C), 14.7 (1 × C); DEPT-135 (100 MHz, DMSO-*d*_6_) δ: 137.2, 129.8, 128.5, 119.3, 114.3, 63.4, 43.7, 40.9, 27.2, 14.7; LRMS [ESI]^+^: *m*/*z* calcd for C_21_H_21_N_7_O [M − H]^−^ 386.2; found 386.3.

#### 3.3.11. 2-((3-(1H-Imidazol-1-yl)propyl)amino)-6-amino-4-(4-(isopropyle)phenyl)pyridine-3,5-dicarbonitrile (**4k**)

^1^H NMR (400 MHz, DMSO-*d*_6_) δ: 8.10 (s, 2H, NH_2_), 7.65 (s, 1H, imidazole), 7.41–7.37 (m, 4H, Ar-H), 7.19 (s, 1H, imidazole), 6.89 (s, 1H, imidazole), 4.17 (br. s, 2H, NH), 4.07 (t, *J =* 7.3 Hz, 2H), 2.96 (sep, *J =* 6.9 Hz, 1H, CH), 2.06 (quint, *J =* 7.6 Hz, 2H, CH_2_), 1.24 (d, *J =* 6.9 Hz, 6H, 2 × CH_3_); ^13^C NMR (100 MHz, DMSO-*d*_6_) δ: 156.3 (1 × C), 150.5 (1 × C), 137.2 (2 × C), 132.5 (1 × C), 128.5 (2 × C), 128.1 (3 × C), 126.5 (3 × C), 119.3 (2 × C), 117.3 (1 × C), 43.7 (1 × C), 41.0 (1 × C), 40.5 (1 × C), 39.0 (1 × C), 33.4 (1 × C), 27.1 (1 × C), 23.7 (1 × C); DEPT-135 (100 MHz, DMSO-*d*_6_) δ: 137.2, 128.5, 128.1, 126.5, 119.3, 43.7, 41.0, 40.5, 33.4, 27.1, 23.7; LRMS [ESI]^+^: *m*/*z* calcd for C_22_H_23_N_7_ [M + H]^+^ 384.2; found 384.3.

### 3.4. Alkaline Phosphatase Inhibition Assay

The newly synthesized compounds were tested against human tissue nonspecific alkaline phosphatase (*h*-TNAP). The compounds were initially tested against the *h*-TNAP enzymes at 1 mM concentration. The assay buffer (pH 9.5) constituted of 50% glycerol, followed by tris-hydrochloride (50.0 mM), MgCl_2_ (5.0 mM), and ZnCl_2_ (0.1 mM) in the solution. The enzyme substrate p-nitrophenyl phosphate (p-NPP) and all the required test compounds were diluted to the required concentration by a dilution buffer method. In a 96-well plate, the assay was started by adding 70 µL of assay buffer followed by 10.0 µL of the test compound (from 1 mM), 10.0 µL of the enzyme (0.5 µL/mL) and then subsequently incubating at 37 °C for 10 min. The reaction was initiated with the addition of 10.0 µL of substrate (5.0 mM p-NPP) into each well and incubated for an additional 30 min. After the second incubation at 405 nm, the change in the absorbance of the released p-nitrophenolate was monitored using a 96-well microplate reader (Bio-TekELx 800TM, Instruments, Inc., Stevens Creek Blvd. Santa Clara, CA 95051 USA). The percent inhibition was calculated for the standard drug (levamisole) and each compound; compounds having more than 50% inhibition were selected for a further determination of the IC_50_ values following the same procedure adopted in the initial screening, repeated in a triplicate [69].

Percent inhibition was calculated by the formula:% Inhibition = [100 − (abs of test comp/abs of control) × 100]

IC_50_ values of the compounds exhibiting greater than 50% inhibition at 1.0 mM were calculated using GraphPad software, San Diego, CA, USA.

### 3.5. MTT Cell Viability Assay

The anti-proliferative activity of the synthesized compounds was determined in MCF-7 and HeLa cells by an MTT cell viability assay, as described earlier [69]. Briefly, we seeded 1 × 10^4^ cells/well in a sterile 96-well culture microtiter plate and incubated them in a 5% CO_2_ incubator at 37 °C for 24 h. The compounds were initially tested at 100 µM/mL; compounds having more than 50% inhibition were tested at dilutions (150–12.5 µM/mL) to calculate the IC_50_ values while 1% DMSO was used as control. After 24 h, the cells were treated with 0.2 mg/mL of MTT reagent for 4 h. We added 10% acidified SDS (sodium dodecyl sulfate) solubilizing solution in propanol (1:1) to solubilize the formazan crystals. After 30 min at 37 °C, the plate was placed on a gyratory shaker to completely dissolve the formazan crystals. The plate was placed on a microplate reader to measure the optical density at 570 nm. The IC_50_ values were calculated from three independent experiments as previously reported [65].

### 3.6. Microscopic Analysis of Cytotoxic and Pro-Apoptotic Effect

A microscopic analysis of most active compound **4d** was carried out according to the methods of a published study with marginal modifications [70]. Cells at a density of 2 × 10^5^ cells/well/mL were grown on round coverslips in a 24-well plate and kept in an incubator with humid 5% CO_2_ and 95% air at 37 °C. After an overnight incubation, the culture medium was replaced with a fresh medium and treated with most active compound at the IC_50_ and 2 × IC_50_ values. After the aspiration of the culture medium from each well, cells were washed with pre-sterilized PBS (phosphate buffer saline). After a fixation with 4% formalin in 0.1% *v*/*v* of Triton X-100 for 5–10 min, cells were washed again with PBS, and treated with 10.0 µL (0.01 mg/mL) of 4′,6-diamidino-2-phenylindole (DAPI) or propidium iodide (PI) dye for 10 min. Cells were gently washed with PBS to remove excess dye, and images were captured using excitation/emission filters at 493/632 and 350/460 nm for the cells treated with PI (20X magnification) and DAPI at 40× magnification for the compound (**4d**), respectively, with the help of a fluorescence microscope (Nikon ECLIPSE Ni–U).

### 3.7. Microscopic Evaluation of ROS Generation

The production of reactive oxygen species (ROS) in compound **4d**-treated HeLa cells was observed by applying a published protocol with some modifications [71]. Cells at a density of 2 × 10^5^ cells/well/mL were grown on round coverslips in a 24-well plate and kept in an incubator having humid 5% CO_2_ and 95% air at 37 °C. After an overnight incubation, the culture medium was replaced with a fresh medium and treated with the most active compound at the IC_50_ and 2 × IC_50_ values. After the aspiration of the culture medium from each well, cells were washed with pre-sterilized PBS (phosphate buffer saline). After fixation with 4% formalin in 0.1% *v*/*v* of Triton X-100 for 5–10 min, cells were again washed with PBS and treated with 10 µL (0.01 mg/mL) of dichlorofluorescein diacetate (H_2_DCF-DA) dye for 10 min. Cells were gently washed with PBS to remove excess dye, and images were captured at a magnification of 20× using excitation/emission filters at 488/530 nm with the help of a fluorescence microscope (Nikon ECLIPSE Ni–U).

### 3.8. Cell Cycle Analysis

An arrest in the cell cycle progression was observed in MCF-7 cells after treatment with compound **4d** and propidium iodide (PI). PI was used as a fluorescent probe by flow cytometry through small modifications in a published method [72]. The difference in DNA content between the G_0_/G_1_, S (synthesis), and G_2_/M phases can be compared with a control sample. The assay was conducted by seeding 3 × 10^5^ cells/well and incubating them for 24 h in a CO_2_ incubator at 37 °C. The incubated cells were treated with IC_50_ and 2 × IC_50_ concentrations of potent sample for 24 h. After harvesting the cells as directed, these cells were centrifuged at 3000 rpm for 5 min and washed with sterile PBS. In order to lessen agglutination during fixation, cells were fixed with 70% ethanol drop-wise in continuous vortexing. After fixation, the cells were placed in a −20 °C environment for 1–2 h. Centrifugation was repeated to collect cell pellets after complete fixation. Once re-suspended, the pellet was diluted with 20 µg/mL of propidium iodide (PI), 0.1 *v*/*v* of triton X-100, and 50 µg/mL of RNse. Samples were placed in the dark for 30 min. Then samples were made to run using a BD Accuri C6 flow cytometer, and 10,000 events were taken. The obtained results were analyzed by using the BD Accuri^TM^ C6 and GraphPad Prism 5.01 software.

### 3.9. Molecular Docking of DNA with Compound ***4d***

The docking study was performed to examine the interactions of compound **4d** with DNA (PDB: 1BNA). This study was carried out by using the Autodock Vina program. [73] Prior to docking, the DNA was prepared by removing water molecules. Autodock Tools [74] were used to add charges and autodock types to convert DNA and ligands into pdbqt formats. The grid box size of 121 × 121 × 121 Å, centered at X = 14.80, Y = 21.00, and Z = 8.90, was defined to ensure the unrestricted movement of ligands and cover the entire DNA. Other parameters were set as follows: number of maximum binding modes = 50; exhaustiveness value = 32; and maximum energy difference between the worst and best binding mode = 4 Kcal/mol, respectively.

### 3.10. Fluorescence Emission Studies

A stock solution (100 µM) of **4d** with a volume of 20.0 mL was prepared in DMF/H_2_O (1:9, *v*/*v*). A 100 µM stock solution of **4d** was diluted to 10 µM in in the final aqueous solution (1000 µL). Aside from this, different contents of DNA, i.e., 10 µg/mL, 20 µg/mL, 30 µg/mL, 35 µg/mL, 40 µg/mL, 45 µg/mL, and 50 µg/mL, were mixed with compound **4d** in the final solution (1000 µL). Each prepared sample was subjected to a spectrofluorometer to record the fluorescence emission spectra at an excitation wavelength (λ_exc_) of 280 nm, maintaining slit widths at 2/2.

## 4. Conclusions

The multi-component reaction protocol provided structurally diverse analogs of dihydropyridine and pyridine in good yields under convenient reaction conditions. These highly functionalized dihydropyridine and pyridine pharmacophores exhibited remarkable inhibitory activity against human tissue nonspecific alkaline phosphatase (*h*-TNAP), with IC_50_ values ranging from 0.49 ± 0.025 to 8.8 ± 0.53 µM. The structure–activity relationship (SAR) and molecular docking studies identified hydrogen bonding, hydrophobic, and π–π stacking interactions with the amino acids. The direct binding of the Zn^2+^ and amino acid residues of the active pocket of the APs of the enzyme with various functional groups of pharmacophores played a critical role in enzyme inhibition. Docking study data have a close resemblance with in vitro *h*-TNAP inhibition. A cytotoxic screening of the synthesized compounds showed good activity of compound **4d** against both HeLa and MCF-7 cell lines. Moreover, we noticed that compound **4d** produced a significant amount of ROS and halted the cell cycle at the G1 phase by inhibiting cyclin-dependent kinase 4 and 6 (CDK4/6). The DNA damage, mitochondrial dysfunction, and shrinkage of both the cell membrane and nuclear membrane were observed through high-resolution microscopy. Furthermore, a fluorescence titration experiment confirmed the binding interaction of the model compound **4d** with DNA. As a whole, 2-naphthyl-substituted dihydropyridine **4d** and related analogs endowed peculiar and unique biological characteristics. This research may offer a promising model for new anticancer drug development in the future.

## Data Availability

Mandatory data will be available on request from corresponding author.

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
