# Peer review of "Design, Synthesis, and Biological Evaluation of Novel Dihydropyridine and Pyridine Analogs as Potent Human Tissue Nonspecific Alkaline Phosphatase Inhibitors with Anticancer Activity: ROS and DNA Damage-Induced Apoptosis"

_molecules, 2022, doi:10.3390/molecules27196235_

Round 1

Reviewer 1 Report

This research paper was focused on the design, synthesis and biological evaluation of dihydropyridines and pyridines as inhibitors of human tissue non-specific alkaline phosphatase (h-TNAP) enzyme. The anticancer activities of these compounds were screened, and the most potent compound 4d was studied for its multiple mechanisms of action. This paper is of importance to the discovery of h-TNAP inhibitors as well as its use in anticancer drug development. The organization of this manuscript should be significantly improved, and some key issues also should be addressed before its publication on Molecules.

Major points:

1. The Introduction section should be strongly abbreviated (just providing the research background, research design/rationale, and the scope of current study). The other related contents may be moved to the discussion section.

2. In the Introduction section, the authors are required to introduce the current TNAP inhibitors used as anticancer drugs/compounds.

3. The design rationales in the Introduction section is not clear. The authors took pyridine/dihydropyridine as the core structures from the anticancer drugs and installed the diverse structures to the pyridine/dihydropyridine cores, and then got several compounds with more enzymatic potency than standard drug. It is kind of weird (All the structure parts seem random, but the activities are significantly potent compared with the standard drug), and the authors are required to discuss more about it.

4. The binding affinity (or the other secondary assays, e.g., in vitro pulldown, thermal shift assay, etc) of the most potent inhibitor 4d is required to validate the results of enzymatic assay.

5. The docking scores/energies are required to be provided to support the results and conclusions in this manuscript.

6. In the cellular toxicity study, the standard drug with the same mechanism is required to be tested as a comparison to the newly synthesized compounds.

7. The toxicity of the most potent 4d is required to be evaluated in normal mammalian cells.

Minor points:

1. In Fig 5, the residue labels (font, size) should be consistent.

2. In Fig 6 and 7, the ‘IC50’ is suggested to be labeled before each value to avoid the misunderstanding of these values.

3. In Fig 8, the x-axis did not present correctly. It should be Log (concentration) value.

4. In Fig 9 and 10, the scale bar should be provided.

5. In Fig 12, the resolution of the labels on the right panel is too low, and the authors are required to adjust it.

6. In Fig 13, the R2 calculation on the right panel is incorrect.

Reviewer 2 Report

The manuscript entitled “Design, Synthesis and Biological Evaluation of Novel Dihydropyridine and Pyridine Analogues as Potent Human Tissue Nonspecific Alkaline Phosphatase Inhibitors with Anticancer Activity: ROS and DNA Damage-Induced Apoptosis” by Khan et al., reports design of  dihydropyridines and pyridine as  tissue non-specific alkaline phosphatase inhibitors for developing anticancer agents.

The manuscript reports a lot of things which are unnecessary. To begin with the introduction appears like a review (badly written), where data has been dumped for different aspects without a running theme to build up strong rationale. The treatment of synthesis appears like a methodology paper. However, that is a fallacy too. Synthesis of dihydropyridines from malononitrile have been previously reported. A google search will show several references. The authors have mentioned (Figures S1‒S28), but the supplementary file is missing. Figure 1 and 3 do not provide any new or valuable insight. Figure 2, choice of colour is strange. Figure 4, most of these drugs have nicotinic acid, nicotinamide and isoquinolones. The figure does not justify the claim that pyridine or dihydropyridine are active pharmacophore. If this is not sought by authors then this becomes redundant. Figure 6 and 7 are occupying a lot of space, instead of figures it could have been a small table. Further table 2, says IC50 values and give some values in percentage inhibition. Figure 8 creates questions on the IC50 values themselves, 4e, 4g and 4j. It seems that they were force fitted. That’s not how these graphs shape up.

Given the quality of content and the treatment and written language, I recommend to reject this paper.

Reviewer 3 Report

The manuscript entitled “Design, Synthesis and Biological Evaluation of Novel dihydropyridine and Pyridine Analogues as Potent Human 3 Tissue Nonspecific Alkaline Phosphatase Inhibitors with 4 Anticancer Activity: ROS and DNA Damage-Induced Apoptosis” may be acceptable in Molecules journal after following corrections are made. Recommend for publication after minor revision.

Comment 1: Authors have done nice piece of work on Novel Dihydropyridine and Pyridine Analogues. This article will be useful to researchers to carryout detail study on mechanism of Human 3 Tissue Nonspecific Alkaline Phosphatase that cause cancer and followed by its inhibition using novel heterocycles. However, manuscript has to fail to address inhibition of cancer-causing others pathways or enzymes such as kinase other than CDK.

Comment 2: Because many compounds having amines functional groups have been for known their toxicity, please include toxicity level for the same. Ref: https://doi.org/10.1002/jsfa.10928

Comment 3: Synthesized compounds structurally similar to apatinib, tyrosine kinase inhibitor. If authors have tested these compounds for tyrosine kinase inhibition, please include.

Comment 4: Under similar conditions, authors obtained pyridine analogs (include N1 and C2 pyridines). Justify the reason. Mentioned the references for the same.

Comment 5: Compounds 4a, 4b displayed 1H-NMR peak at 3.96 (s, 1H, CH), 3.94 (s, 1H, CH). Cross check with literature value.

Comment 6: compound 4j and 4k shows peak in mass spectra at [ESI]+: m/z calcd for C21H21N7O [M+H]+ 386.2; Found 388.3 and LRMS [ESI]+ : m/z calcd for C22H23N7 [M+H]+ 384.2; Found 386.2 Cross check with calculated value.

Comment 7: Authors have mentioned in their docking study that compound 4d has displayed binding affinity with His324 and His437 amino acid residue and its binding leads to increased anticancer activity in similar fashion as reported in reference [55]. Could you please add scientific reference that has shown that binding with His324 and His437 amino acid residue leads to potent anticancer activity.

Comment 8: The authors have to make the necessary corrections for typographic and grammatical mistakes.

Reviewer 4 Report

The manuscript could be accepted for publication in molecules However, the manuscript contain some flaws that should be addressed before recommending it for publication.

My comments and suggestions:

# The stability of the tested inhibitors  under biological conditions should be reported and addressed in the manuscript.

# The authors should show evidence for the sufficient purity of the synthesized  inhibitors such as purity check by HPLC.

# The authors reported that, the synthesized inhibitors have affinity toward DNA through  binding with DNA minor groove as it predicated by molecular docking study. Therefore, the authors should elucidate if the new inhibitors  show cancer cell selectivity, by simultaneously testing  against non-cancerous cell lines (e.g., HLF (human lung fibroblasts).

# The conclusion is too long and should be shortend.

# # The authors cited many references and most of them are a bit old. They should either update the old ones or omit some of them.

Round 2

Reviewer 1 Report

After the authors’ revision, the quality of this paper was significantly improved, and could reach the required quality standard for Molecules in my opinion. I suggest accepting it without further revision.

Reviewer 2 Report

It is a much improved version. The authors have answered my concerns.

Reviewer 4 Report

Comments to Author: All of the concerns are addressed in the revised version and I can see significant improvement in the manuscript and I have no further comments. In my opinion the revised manuscript is now suitable for publication in its current form in Molecules.